# Zn^2+^ Aggravates Tau Aggregation and Neurotoxicity

**DOI:** 10.3390/ijms20030487

**Published:** 2019-01-23

**Authors:** Xuexia Li, Xiubo Du, Jiazuan Ni

**Affiliations:** 1Changchun Institute of Applied Chemistry, Chinese Academy of Sciences, Changchun 130022, China; xxli@ciac.ac.cn; 2School of Applied Chemistry and Engineering, University of Science and Technology of China, Hefei 230026, China; 3College of Life Sciences and Oceanography, Shenzhen Key Laboratory of Microbial Genetic Engineering, Shenzhen University, Shenzhen 518060, China

**Keywords:** tau-R3, Zn^2+^, aggregation, neurotoxicity

## Abstract

Alzheimer’s disease (AD) is a neurodegenerative disease with high morbidity that has received extensive attention. However, its pathogenesis has not yet been completely elucidated. It is mainly related to β-amyloid protein deposition, the hyperphosphorylation of tau protein, and the loss of neurons. The main function of tau is to assemble tubulin into stable microtubules. Under pathological conditions, tau is hyperphosphorylated, which is the major component of neurofibrillary tangles (NFT) in AD. There is considerable evidence showing that the dyshomeostasis of Zn^2+^ is closely related to the development of AD. Herein, by using the third repeat unit of the microtubule-binding domain of tau (tau-R3), we investigated the effect of Zn^2+^ on the aggregation and neurotoxicity of tau. Experimental results showed that tau-R3 probably bound Zn^2+^ via its Cys residue with moderate affinity (association constant (Ka) = 6.82 ± 0.29 × 10^4^ M^−1^). Zn^2+^ accelerated tau-R3 aggregation and promoted tau-R3 to form short fibrils and oligomers. Compared with tau-R3, Zn^2+^-tau-R3 aggregates were more toxic to Neuro-2A (N2A) cells and induced N2A cells to produce higher levels of reactive oxygen species (ROS). The dendrites and axons of Zn^2+^-tau-R3-treated neurons became fewer and shorter, resulting in a large number of neuronal deaths. In addition, both tau-R3 and Zn^2+^-tau-R3 aggregates were found to be taken up by N2A cells, and more Zn^2+^-tau-R3 entered the cells compared with tau-R3. Our data demonstrated that Zn^2+^ can aggravate tau-R3 aggregation and neurotoxicity, providing clues to understand the relationship between Zn^2+^ dyshomeostasis and the etiology of Alzheimer’s disease.

## 1. Introduction

Alzheimer’s disease (AD) was first found and named by Alois Alzheimer in 1906 [1]; it accounts for 50–60% of all forms of dementia and is characterized by the extracellular deposition of amyloid plaques consisting of amyloid-β (Aβ) peptides [2], abnormally hyperphosphorylated tau (p-tau) containing intracellular neurofibrillary tangles (NFTs) [3], and neuronal cell death. The 2018 World Alzheimer Report revealed that there were 50 million people worldwide living with AD in 2015, and this number will reach 152 million in 2050 [4]. AD has a huge impact on the economy, but the exact cause is not yet clear.

Tau is a microtubule-associated protein (MAP) [5,6] that mainly presents in the axons of nerve cells, and binds to tubulin to form the core of early assembling microtubules. The abnormal aggregation of tau proteins forms insoluble paired helix filaments (PHF) that cause nerve fiber tangles, which are one of the main pathological features of AD. Therefore, the molecular mechanism of the abnormal aggregation of tau protein has become the key to the study of AD pathology. In the human central nervous system, there are six tau isoforms obtained by the translation of mRNAs produced by different cleavage of the same gene [7]. Tau protein has two major domains: the overhanging domain at the N-terminus, and the microtubule-binding domain at the C-terminus. The latter determines the biological function of tau. As shown in Figure 1, the microtubule-binding domain of tau protein contains three (R1, R3, and R4) or four repeats (R1–R4), with each consisting of 31 or 32 amino acid residues [8]. The repeat domains not only participate in microtubule stabilization, but also form the core of the PHF [8,9,10]. R2 and R3 contain a hexapeptide (^275^VQIINK^280^ or ^306^VQIVYK^311^) motif, and exhibited similar properties to the full-length tau protein in vitro experiments, which also makes them the best polypeptide fragments to mimic tau aggregation. For example, both of them can self-aggregate to form PHFs and NFTs in the presence of an inducer [11], while only R3 is present in all the six tau isoforms. R3 also has the smallest self-aggregation concentration and the highest self-aggregation rate [12], so the R3 fragment was used in this study to simulate the aggregation behavior of full-length tau protein.

The dynamic imbalance of metal ions such as Cu^2+^, Fe^3+^, Zn^2+^, and Ca^2+^ in the brain is closely related to the pathogenic mechanism of AD [13,14,15]. Under normal conditions, the concentration of free metal ions in the brain is very low, and does not affect neurological function. Meanwhile, the brain tissue of AD patients contains large amounts of transition metal ions, such as Cu^2+^, Fe^3+^, and Zn^2+^ [16]. These metal ions are closely related to the aggregation of Aβ, the phosphorylation of tau, the generation of reactive oxygen species (ROS), and neuron death [17,18]. It is reported that Cu^2+^ is an inducer of self-assembly of the R3 peptide, and makes the R3 peptide form a structure similar to PHF [19]. Our previous studies have found that Cu^+^ and Cu^2+^ ions induced the aggregation and neurotoxicity of tau-R2, which bound 0.44 Cu^2+^ and 0.34 Cu^+^ per monomer with dissociation constants of 1.1 nM and 0.2 pM, respectively [20].

As an essential trace element, Zn^2+^ is involved in human growth and development, immune regulation, protein and nucleic acid synthesis, and other physiological activities [21,22]. Appropriate Zn^2+^ concentration plays a crucial role in nervous system development and the differentiation of neural stem cells. The imbalance of Zn^2+^ homeostasis is considered to be an important influencing factor for some neurodegenerative diseases [23]. It was found that the mRNA levels of Zn^2+^ transporters, including LIV1, ZIP1, ZnT1, ZnT4, and ZnT6 were increased in the cortex of postmortem brain tissues from AD patients, increasing the likelihood of interactions between Zn^2+^ and Aβ or tau protein in the brains [24]. There is increasing evidence suggesting that Zn^2+^ can combine Aβ and tau, which is crucially involved in the pathogenesis of AD. Zn^2+^ levels were higher in amyloid plaques in AD [16], and excess Zn^2+^ exposure could be a risk factor for AD pathological processes [25]. Huang et al. found that Arg13, His6, and His14 residues provided the primary binding sites for Zn^2+^ in the rat Aβ_1-28_ peptide, and the proper binding of Zn^2+^ induced the peptide to adopt a more stable conformation [26]. Mo et al. demonstrated that low micromolar Zn^2+^ accelerated the fibrillization of human tau protein via bridging Cys-291 and Cys-322 in physiological reducing conditions [27]. Sun et al. found that synaptically released Zn^2+^ promoted tau hyperphosphorylation through protein phosphatase 2A (PP2A) inhibition [28]. Thus, abnormal Zn^2+^ homeostasis is believed to be a contributing factor leading to tau aggregation, and the alteration of Zn^2+^ homeostasis is a potential therapeutic strategy for AD.

In addition, levels of manganese, molybdenum, and iron were also changed significantly among Alzheimer disease, mild cognitive impairment, subjective memory complaint, and healthy subjects [29]. Thus, in the present study, by using isothermal titration calorimetry (ITC) and UV-vis spectroscopy, the thermodynamic properties of tau-R3 binding Mn^2+^, Mo^5+^, Fe^3+^, and Zn^2+^ were characterized. By thioflavin T (ThT) fluorescence assay and circular dichroism spectroscopy (CD), the aggregation kinetics and morphology of tau-R3 in the presence or absence of metal ions were determined. The viabilities and the intracellular ROS levels of N2A cells exposed to different tau-R3 aggregates were measured by CCK-8 assay and flow cytometry. In addition, we demonstrated that exogenous tau-R3 and Zn^2+^-tau-R3 aggregates could enter cells, and the dendrites and axons of Zn^2+^-tau-R3-treated neurons become fewer and shorter, resulting in a large number of neuronal deaths. Our findings linked Zn^2+^-induced tau aggregation and neurotoxicity to the pathogenesis of AD.

## 2. Results

### 2.1. Tau-R3 Bound Zn^2+^ with Moderate Affinity

To investigate the binding of metal ions to tau-R3, the peptide (100 μM) was titrated with different metal cations (two mM) in their chloride forms, and the calorimetric changes were monitored using iTC200, which provides a direct route to the thermodynamic characterization of noncovalent equilibrium interactions. ITC profiles for the titration of Mn^2+^, Mo^5+^, or Fe^3+^ to tau-R3 at 25 °C suggested that there is no specific binding between tau-R3 and these metal cations (Appendix A). However, the titration of Zn^2+^ into the tau-R3 peptide resulted in large exothermic peaks, which eventually diminished to only the heat of the dilution after nine injections (Figure 2A). The calorimetric data were best fit to a model assuming a single set of identical binding sites, which approximated the association constant (Ka) to be 6.82 ± 0.29 × 10^4^ M^−1^, and the binding stoichiometry (*n*) to be 0.43 ± 0.01 (Figure 2A). The ΔH and ΔS were estimated to be −16.58 ± 0.33 kcal mol^−1^ and −33.5 cal/mol/deg, respectively. Therefore, the binding of Zn^2+^ to tau-R3 is an entropy-driven, but not enthalpy-driven reaction.

The binding of Zn^2+^ and tau-R3 was also monitored by UV-vis spectrometry, and the results are shown in Figure 2B. With the titration of Zn^2+^, the absorption spectrum of tau-R3 showed a peak at 218 nm assigned to an S–Zn charge transfer [30], indicating that the single cysteine in tau-R3 contributed a ligand coordinating Zn^2+^. In agreement with this, Mo et al. reported that Zn^2+^ bound to human tau via Cys-291 and Cys-322, which are located in the R2 and R3 fragments of tau, respectively [27].

### 2.2. Zn^2+^ Accelerated the Fibrillization of Tau-R3 In Vitro

The fibrillization of tau-R3 incubated alone or with different cations (Mn^2+^, Mo^5+^, Fe^3+^, or Zn^2+^) was examined by a ThT fluorescence assay [31,32]. ThT specifically binds to the β-sheet structure rapidly, and its fluorescence intensity at 480 nm positively correlates with the content of fibers. Heparin was added to induce the fibrillization of tau-R3. As shown in Figure 3, the kinetic curves of ThT fluorescence intensity at 480 nm for tau-R3 fibrillization consisted of three phases: the lag phase, the exponential phase, and the stable phase, which fit well with the sigmoidal function Boltzmann equations. Mn^2+^, Mo^5+^, and Fe^3+^ all significantly promoted the fibrillization of tau-R3, which was reflected by shorter lag phases and much higher ThT fluorescence intensities. In the presence of Mn^2+^, after incubation at 37 °C for 30 h, the highest ThT fluorescence intensity was approximately 2.2-fold that of tau-R3 incubated alone. Interestingly, Zn^2+^ slightly increased the fibers content of tau-R3, but it markedly accelerated the fibrillization of tau-R3, with the *t*_50_ (the time to 50% fibrillization) reduced from 15 h (tau-R3 alone) to six h.

To better understand the effects of different metal ions on tau-R3 aggregation, we analyzed the secondary structures of tau-R3 via CD spectra. A far-UV spectroscopy signal is generated due to asymmetric structures around the chromophores in the peptide chain of the protein, such as α-helix, β-sheet, and irregular curl structures, so the CD signal in this region can provide protein secondary structure information. The CD spectra of tau-R3 incubated at 37 °C alone or with different metal ions from 0 to 24 h were recorded and shown in Figure 4. At zero h, tau-R3 mainly adopted a random coil conformation, which was characterized by a negative peak near 198 nm (Figure 4A). All of the metal ions that were examined promoted the conversion of tau-R3 from the random coil to the β-sheet. When incubated with Mn^2+^, Mo^5+^, or Fe^3+^ for 3 h, the CD spectra of tau-R3 showed a mixed structure of random coils and β-sheets. While in the presence of Zn^2+^, almost all of the tau-R3 turned to adopt the β-sheet confirmation after the 3-h incubation, which was characterized by a negative peak at 216 nm and a strong positive peak at 196 nm (Figure 4B). Tau-R3 completely converted from random coil to β-sheet in 24 h, whether it was incubated alone or co-incubated with metal ions (Figure 4C). Ellipticities at 198 nm, 216 nm, and 196 nm were plotted against the incubation time (Figure 4D,E), which showed a decrease in random coil content and an increase in the β-sheet content. Consistent with the ThT fluorescence assay, the CD spectra demonstrated that Zn^2+^ markedly accelerated the aggregation of tau-R3.

### 2.3. Zn^2+^ Changed the Morphology of Tau Aggregates

The morphologies of tau-R3 aggregates incubated with or without cations (Mn^2+^, Mo^5+^, Fe^3+^, and Zn^2+^) at 37 °C for 24 h were further investigated by atomic force microscope (AFM). As shown in Figure 5, when incubated alone, tau-R3 aggregates were mainly composed of long filaments (Figure 5A). Mn^2+^, Mo^5+^, and Fe^3+^ did not affect the morphologies of tau-R3 aggregates (Figure 5B–D), although they promoted the fibrillization of tau-R3, as revealed by ThT fluorescence and CD spectra. Interestingly, when co-incubated with Zn^2+^, tau-R3 aggregates formed much shorter filaments (indicated with red arrows) and many oligomers (indicated with blue arrows) (Figure 5E).

### 2.4. Zn^2+^ Aggravates Tau-R3-Mediated Toxicity and ROS Induction in Nerve Cells

Many studies have shown that tau oligomers are more neurotoxic than monomers and NFTs [33]. We measured the toxicities of tau-R3 aggregates prepared in the presence or absence of Zn^2+^ to the mouse Neuro-2A (N2A) neuroblastoma cells using the Cell-Counting Kit-8 Assay. As shown in Figure 6A, either tau-R3 aggregates (15 μM) or Zn^2+^ (7 μM) alone had limited toxicity to N2A cells, with viabilities of 90.6% and 87.4% of the control group, respectively. When the cells were treated with Zn^2+^-induced tau-R3 aggregates (15 μM of tau-R3 incubated with seven μM of Zn^2+^ for 24 h before adding to the cell culture), their viabilities were decreased to 53.1% of the control group. Therefore, Zn^2+^ not only promoted the aggregation of tau-R3, but also aggravated its toxicities to the neuronal cells. Studies have shown that tau oligomers are the most toxic species among the different aggregation forms of tau [34]. The formation of tau-R3 oligomers induced by Zn^2+^ may be one of the reasons that Zn^2+^-tau-R3 aggregates were more toxic.

To further understand the mechanism of the Zn^2+^-induced neurotoxicity of tau-R3, the total ROS levels in N2A cells treated with tau-R3 aggregates, Zn^2+^, or Zn^2+^-tau-R3 aggregates were detected with the fluorescent probe 2,7-Dichlorodi-hydrofluorescein diacetate (DCFH-DA) and quantified with flow cytometry. As shown in Figure 6B, treatment with tau-R3 aggregates or Zn^2+^ alone for 24 h promoted the ROS production in N2A cells by factors of 1.37-fold and 1.26-fold, respectively. Meanwhile, the Zn^2+^–R3 aggregates dramatically stimulated the generation of ROS, with a 1.52-fold increase relative to the control group (Figure 6B, lane 4). It has been demonstrated that metal ions including iron, copper, and zinc promote the aggregation of Aβ and tau, and induce ROS production and oxidative stress [20,35,36]. Excessive ROS led to lipid peroxidation, protein, DNA and RNA oxidation, and mitochondrial dysfunction, and oxidative stress is a prominent and early feature in the pathogenesis of neuronal damage in AD [37].

### 2.5. The Toxicity of Tau-R3 to Primary Neurons

Microtubule-associated protein 2 (MAP-2) is a neuron-specific cytoskeletal protein that acts as a marker for nerve cells and can indicate the growth state of neurons. Thus, we performed immunofluorescence analyses by staining for MAP-2 in primary neuronal cultures (Figure 7) to directly view the growth status of hippocampal neurons (Figure 7A–D) and cortical neurons (Figure 7E–H) upon the treatment of Zn^2+^ and/or tau-R3 for 12 h. As shown in Figure 7A and Figure 8E, neurons maintained in normal media were rich in protrusions and branches, which make the neurons connect with each other to support normal synaptic functions. When cultured with tau-R3 aggregates (Figure 7B,F) or Zn^2+^ (Figure 7C,G), neuronal dendrites and axons were damaged. While incubated with Zn^2+^-tau-R3 (Figure 7D,H), the damage was exacerbated significantly. The dendrites and axons of neurons almost disappeared, resulting in the disappearance of synaptic structures of connected neurons.

### 2.6. Subcellular Distribution of Tau-R3

The above results indicate that the exogenous Zn^2+^-tau-R3 is more toxic to N2A cell lines and nerve cells. To explore the mechanism, we checked whether the different tau-R3 aggregates could enter the cells. Thus, we added the tau-R3 protein labled with fluoresceine isothiocyanate (tau-R3–FITC) aggregates prepared in the presence or absence of Zn^2+^ to the N2A cells, which were stained with organelle-specific fluorescent red dyes for plasma membrane, endoplasmic reticulum, or Golgi. Confocal analysis showed that both tau-R3 aggregates and Zn^2+^-tau-R3 oligomers entered the N2A cells, and were mainly distributed throughout the cytoplasm, including the plasma membrane (Figure 8). We also found that more Zn^2+^-tau-R3 entered the cells compared with tau-R3, which may be one of the reasons that Zn^2+^-tau-R3 was more toxic to neuronal cells.

## 3. Discussion

As one of the most abundant essential elements in the brain, Zn^2+^ is involved in normal brain development and nervous functions, especially in the modulation of synaptic transmission and plasticity [38,39,40]. In AD brains, Zn^2+^ concentrations were elevated in degenerated brain regions, and much higher levels of Zn^2+^ were detected in amyloid plaques and NFT [41]. Impaired Zn^2+^ homeostasis in the brain was linked with the development of AD pathology [42,43,44]. Previously, we and others have reported that Zn^2+^ at micromolar concentrations strongly inhibited the fibrillization of Aβ and promoted it to grow into amorphous aggregates [45]. Here, we first investigated the thermodynamic properties of tau-R3 binding with Zn^2+^ and other metal ions including Mn^2+^, Mo^5+^, and Fe^3+^ by ITC, and found that only the titration of Zn^2+^ into tau-R3 solution resulted in large exothermic peaks. This suggested the specific binding between tau-R3 and Zn^2+^. We found that tau-R3 bound Zn^2+^ probably via its Cys residue with moderate affinity (the dissociation constant was calculated to be approximately 14.7 μM), to form a 1:2 Zn^2+^-R3 complex. The physiological concentration of free Zn^2+^ in cells is between 1–10 nM, but in AD brains, Zn^2+^ is enriched, and its intracellular concentrations can be in the range of 10–300 μM. [46] Therefore, the binding constant obtained in this study is reasonable. Full-length tau has two Cys (C291 and C322) residues, which were located in the R2 and R3 fragments, respectively. Zn^2+^ bound to full-length tau by interacting with C291 and C322, with the stoichiometry value *n* = 1 and binding constant exceeding 10^6^ M^−1^ [47,48]. When Cys291 was mutated to Ala, the binding number and dissociation constant were decreased to 0.44 μM and 9.71 μM, respectively [27], which were much comparable with the values obtained with tau-R3 in this study. Although there is no specific binding between tau-R3 and Mn^2+^, Mo^5+^, or Fe^3+^, these metal ions did promote the fibrillization of tau-R3, as indicated by much higher ThT fluorescence intensities than tau-R3 incubated alone or co-incubated with Zn^2+^ in Figure 3. ThT can specifically bind to the β-sheet structure rapidly, but not to monomeric or oligomeric intermediates. Therefore, the fluorescence intensity at 480 nm positively correlates with the fiber content. Interestingly, Zn^2+^ dramatically accelerated the aggregation of tau-R3, with the *t*_50_ (the time to 50% fibrillization) reduced from 15 h (tau-R3 alone) to six h. In addition, in the presence of Zn^2+^, tau-R3 formed much shorter filaments and many oligomers. Consistently, Zn^2+^ also accelerated the fibrillization of full-length tau and induced tau to form shorter fibrillar aggregates [47,48].

As a microtubule (MT)-associated protein, tau binds to and promotes the assembly of microtubules. However, in diseased conditions, tau is subjected to hyperphosphorylation and detaches from the MTs. The now free tau assembles into small aggregates known as tau oligomers in route of NFT formation. A large body of studies has shown that tau oligomers are the most toxic species among the different aggregation forms of tau [34]. It was demonstrated that brain injection of tau oligomers rather than tau monomers or fibrils led to the synaptic, mitochondrial, and cognitive abnormalities in the mice [49]. In our study, both tau-R3 and Zn^2+^-tau-R3 aggregates could enter cells, and more Zn^2+^-tau-R3 entered the cells compared with tau-R3. Consistent with this, Zn^2+^-tau-R3 aggregates showed much higher toxicities toward N2A cells, with lower cell viabilities and higher intracellular ROS levels. We also tested the toxicities of different tau-R3 aggregates to the hippocampal neuros. As shown in Figure 7, when incubated with tau-R3 aggregates, the neurons showed a very short axon and little dendrite branching. The situation of neurons cultured with Zn^2+^-tau-R3 aggregates was even worse. The neurons were degraded, almost no neurite was observed, and the communication between neurons was absolutely disrupted. We suggest that Zn^2+^ enhanced tau-R3 toxicities at least partially due to the formation of tau oligomers. More noteworthy, studies have discovered that tau oligomers induced endogenous tau to misfold and propagate from affected to unaffected brain regions in mice [33].

Our findings suggest that tau-R3 aggregates had limited toxicities to nerve cells. In the presence of Zn^2+^, the toxicities of tau-R3 were severely enhanced. Considering the concept that tau oligomers are the toxic entities responsible for neurodegeneration in tauopathies and that tau pathology propagation from affected to unaffected brain regions was dependent on tau oligomers but not monomers or fibrils, we propose that the augmented toxicity is at least partially attributed to tau oligomers formation induced by Zn^2+^. Based on our data, we propose a hypothetical model to demonstrate how Zn^2+^ aggravates tau-R3 aggregation and toxicity (Figure 9). Tau-R3 binds Zn^2+^ with moderate affinity and forms the 1:2 Zn^2+^-tau-R3 complex. Zn^2+^ remarkably accelerates the aggregation of tau-R3, and meanwhile promotes the formation of tau oligomers. Some of the Zn^2+^-tau aggregates, especially in the soluble oligomeric form, are then taken up by cells via endocytosis, micropinocytosis, or some unknown pathway. Once entering cells, tau oligomers, together with the extracellular large fibrils, promote ROS production [34]. Our study suggests an involvement of Zn^2+^ in the pathogenesis of AD.

## 4. Materials and Methods

### 4.1. Materials

Tau-R3 (306-336: VQIVYKPVDLSKVTSKCGSLGNIHHKPGGGQ) and tau-R3-FITC were synthesized at China Peptides Co., Ltd. (Shanghai, China). Thioflavin T (ThT) was purchased from Sigma. Heparin (average molecular mass of 12 kDa) was obtained from Aladdin. All of the metal cations used were chloride forms of analytical grade, which were all purchased from Macklin, and Tris-base was purchased from VOVON. Cell culture media was purchased from HyClone and Gibco. The Cell-Counting Kit-8 Assay, Oxygen Species Assay Kit, Hoechst 33342, Dil-Tracker Red, ER-Tracker Red, and Golgi-Tracker Red were all obtained from Beyotime Biotechnology (Beyotime, Nanjing, China). All of the other chemicals that were used were of analytical grade.

### 4.2. Isothermal Titration Calorimetry (ITC)

ITC experiments were carried out at 25.0 °C using the MicroCal iTC-200 microcalorimeter (GE Healthcare, New York, NY, USA), as described previously [50]. Briefly, tau-R3 (100 μM, in 50 mM of Tris-HCl buffer, pH 7.4) was loaded into the sample cell (200 μL), and metals ions (Zn^2+^, Fe^3+^, Al^3+^, or Cr^2+^, 2 mM) in Tris-HCl buffer (50 mM, pH 7.4) were added to the syringe (40 μL). The titration parameters were set as follows: stirring speed, 1000 rpm; titration volume per drop, two μL; duration of each drop, 4 s; titration interval time, 120 s; total number of titrations, 19 times. The resulting data were fitted to a one-site binding model using MicroCal ORIGIN software (ORIGIN 7.0, Northampton, MA, USA) supplied with the instrument. A nonlinear least-squares method was used to obtain the best-fit parameters, including the number of binding sites, *n*; the association constant, *Ka*; and the change of enthalpy, *ΔH*. All of the experiments were performed in triplicate under the same conditions.

### 4.3. UV-Vis Spectroscopy

UV/Vis spectra were monitored on a LAMBDA 25 UV-visible spectrophotometer (Perkin Elmer, Waltham, MA, USA) with a one-cm cuvette at room temperature. Briefly, one μM of tau-R3 peptide in Tris buffer (50 mM, pH 7.4) was titrated with 0.1 mM of ZnCl_2_. The absorption spectra were recorded over the wavelength range of 200–300 nm, and normalized with absorbance at 300 nm.

### 4.4. Monitoring Tau-R3 Fibrillization by Thioflavin T Fluorescence

The one-mM ThT stock solution was freshly prepared in 50 mM of Tris-HCl buffer (pH 7.4) and passed through a 0.45-μm pore size filter to remove insoluble particles prior to use. Tau-R3 (150 μM) was incubated with 20 μM of ThT and 16 μM of heparin in the presence or absence of 10 μM of metal ion solutions. Samples (200 μL) were transferred to 96-well plates and incubated at 37 °C for 31 h. The fluorescence of ThT was excited at 440 nm, and the emission was measured at 480 nm on a fluorescence spectrophotometer (Fluoroskan Ascent FL 4500, Thermo Scientific, Waltham, MA, USA), with the measurement interval of 10 min. Each measurement was run in triplicate or quadruplicate per experiment, and repeated at least three times.

### 4.5. Circular Dichroism Spectroscopy

The effects of metal ions on the secondary structure of tau-R3 were analyzed by CD spectra. Briefly, tau-R3 protein and metal ions were prepared in PB buffer at concentrations of 50 μM and 10 μM, respectively. Tau-R3 was incubated with or without 10 μM of different metal ions in the presence of heparin at 37 °C for zero to 24 h, before their CD spectrum were collected at 25 °C using a J-815 circular dichroism analyzer (JASCO, Tokyo, Japan). Before performing the experiments, the instrument optics and sample chamber were flushed with high-purity N2. The parameter settings were as follows: wavelength range, 190–250 nm; scanning speed, 50 nm/min; sensitivity, 50 mdeg; reaction time, 0.25 s; band width, 0.5 nm; number of scans, three. The spectra were corrected for the corresponding buffer and smoothed using a fast Fourier transform filter (FFT). The results that were obtained were plotted using the Origin 8. 0 software (ORIGIN 8.0, Northampton, MA, USA).

### 4.6. Atomic Force Microscope (AFM)

Tau-R3 (50 μmol/L) was incubated with heparin (16 μmol/L) in the presence or absence of metal ions (10 μmol/L) at 37 °C for 24 h before imaging. The mica sheet was secured on the metal substrate sheet with double-sided tape. Five to 10 layers of mica surface were freshly disassociated with transparent glue until the surface of the mica plate was flat and smooth. The 50-fold diluted sample (50 μL) was placed on the flat and smooth mica plate and allowed to stand for 5 min. Then, we drew one mL of ultrapure water and slowly washed the mica flakes. We repeated this twice, and the excess water was absorbed by filter paper. The tablets were sealed and then dried for 12 h, and finally observed by AFM scanning (Bruker Dimension Icon Scanning Probe Microscope, (Bruker, Santa Barbara, CA, USA)).

### 4.7. Cell Culture

Wild-type murine neuroblastoma Neuro-2A cells (N2A/WT) cells were purchased from the Shanghai Institute of Biological Sciences, Chinese Academy of Sciences (Shanghai, China). The cells were cultured in a mixed medium containing 5% fetal bovine serum, 1% penicillin G and streptomycin, 49% dulbecco’s modified eagle medium (DMEM), and 45% Opti-MEM. Tau-R3 (15 μM) was incubated with heparin (16 μM) in the presence or absence of metal ions (seven μM) at 37 °C for 24 h, before being applied to N2A cells for 24 h.

### 4.8. Cell Viability Measurement

N2A cells were cultured on a 96-well cell culture microplate at the density of 5 × 10^3^ cells/mL. After 24 h of incubation with tau-R3 in the presence or absence of Zn^2+^, the medium was removed, and 100 μL of fresh medium containing 10 μL of CCK-8 reagent was added. After incubation for 1 h, the absorbance at 450 nm was measured with a reference wavelength at 650 nm using a spectra MAX 190 microplate reader (Molecular Devices, Sunnyvale, CA, USA). Triplicates were performed throughout the procedures.

### 4.9. Measurement of ROS

N2A cells were cultured on six-well plates at the density of 1 × 10^5^ cell/mL. 2′,7′-Dichlorofluorescin diacetate (DCFH-DA) was diluted with serum-free DMEM to a final concentration of 10 μM. Then, the old medium was removed from the culture dish, and two mL of diluted DCFH-DA was added into each well. Cells were incubated at 37 °C for another 20 min, and then washed three times with serum-free DMEM to fully remove the extracellular DCFH-DA. Cells were harvested and washed three times with prechilled phosphate buffer saline (PBS) and resuspended in PBS. The fluorescence intensity was excited at 488 nm, and the emission was measured at 535 nm by flow cytometry (FACS Calibur, Becton Dickinson, NJ, USA). The results are expressed as fold change compared with the corresponding controls. Assays were run in triplicate.

### 4.10. Neuronal Culture

The hippocampus and cortex of the brain of newborn B6129SF2/J mice were digested with 0.4 mg/mL of papain at 37 °C for 30 min. The digested tissue was suspended, centrifuged, resuspended, counted, and then seeded in a six-well cell culture plate (7 × 100,000 cells/well) pre-coated with 0.1 mg/mL of polylysine. After 4 h, the old medium was changed with raise medium (neurobasal medium with 2% B27 supplement, 0.5 mL of Lglutamine and 50 U/mL of penicillin-streptomycin), and then one-half of the medium was replaced every three days. On day seven, the cultured neurons were treated with tau aggregates (7.5 μM tau-R3, 3.5 μM Zn^2+^, and 1.9 μM heparin) and incubated at 37 °C for 24 h before being added to the neurons) for 12 h.

### 4.11. Immunofluorescence Analysis of Map2 Protein

For immunocytochemistry, cells were fixed in the cell culture dishes with 4% paraformaldehyde in PBS, and then permeabilized with 0.2% Triton X-100 in PBS. Nonspecific binding was blocked with 10% goat serum in PBS for 1 h at room temperature. Primary anti-MAP-2 antibody (1:200, rabbit monoclonal, ProteinTech, Chicago, IL, USA) diluted in blocking solution was applied and incubated at 4 °C overnight followed by three washes with PBS. Secondary antibody conjugated to Dylight-488 (1:1000 dilution, Abcam, Cambridgeshire, England) was then added and incubated at room temperature for 1 h. Cells were washed with PBS three times, and then visualized using a confocal microscope (ZEISS, Jena, Germany).

### 4.12. Cellular Localization of Tau-R3 and Zn^2+^-Tau-R3 in N2A Cells

N2A cells were cultured on laser confocal dishes at the density of 1 × 10^5^ cell/mL at 37 °C for 24 h. Then, the old medium was removed from the culture dish, and two mL of 15 μM of tau-R3-FITC aggregates prepared in the presence or absence of seven μM of Zn^2+^ were added into each well. Cells were incubated at 37 °C for another 24 h, and then washed three times with serum-free DMEM. Hoechst 33342, Dil-Tracker Red, ER-Tracker Red, and Golgi-Tracker Red obtained from Beyotime Biotechnology were then used to stain the nuclei, plasma membrane, endoplasmic reticulum, and Golgi of the N2A cells. The experimental steps were all carried out according to the kit instructions. Then, the cells were visualized using a confocal microscope (ZEISS, Jena, Germany).

### 4.13. Statistical Analysis

The data in all of the panels were expressed as the mean ± S.E.M. *p* values were determined by two-tailed, unpaired t-tests with GraphPad Prism5.0 software (Lo Jolla, CA, USA).

## 5. Conclusions

In the present study, the effects of Zn^2+^ on tau-R3 aggregation and neurotoxicity were investigated through a series of chemical and biological experiments. The combination ratio of Zn^2+^ and tau-R3 is 1:2 with a binding constant of K = (6.82 ± 0.29) × 10^4^ M^−1^ via isothermal titration calorimetry. Through thioflavin T fluorescence, circular dichroism spectroscopy, and TEM, we found that Zn^2+^ accelerated tau-R3 aggregation, decreased the content of irregular curl with the increasing of the β-sheet, and promoted tau-R3 to form parts of shorter, smaller oligomers. Compared with tau-R3, Zn^2+^-tau-R3 aggregates were more toxic to N2A cells and induced N2A cells to produce more ROS. The dendrites and axons of Zn^2+^-tau-R3-treated neurons become fewer and shorter, resulting in a large number of neuronal deaths. Our data demonstrate that Zn^2+^ can aggravate tau-R3 aggregation and neurotoxicity, providing clues to understanding the relationship between Zn^2+^ dyshomeostasis and the etiology of Alzheimer’s disease.

## Figures and Tables

**Figure 1 ijms-20-00487-f001:**
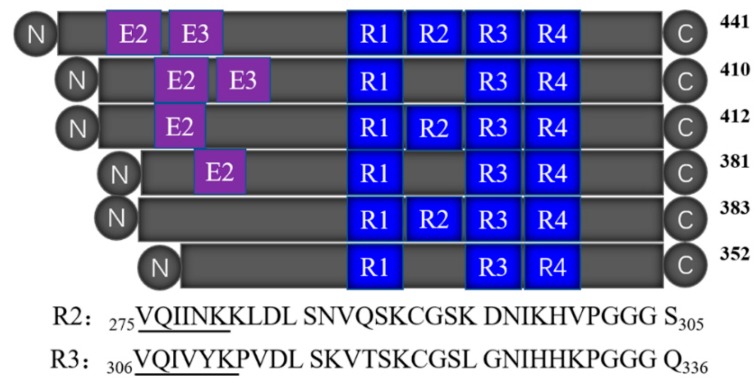
The sequence and location of R2 and R3 in six tau isoforms.

**Figure 2 ijms-20-00487-f002:**
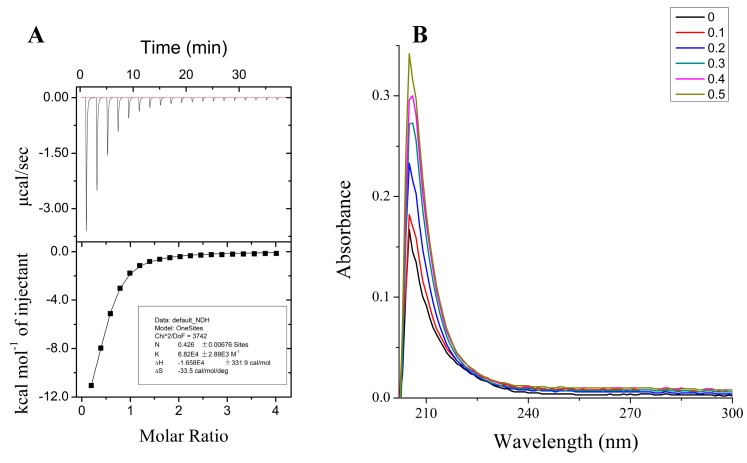
Properties of Zn^2+^ binding to the third repeat unit of the microtubule-binding domain of tau (tau-R3). (**A**) Calorimetric titration of Zn^2+^ to tau-R3. The top panel represents the raw data for sequential two-μL injections of Zn^2+^ into tau-R3. The bottom panel shows the plot of the heat evolved (kilocalories) per mole of Zn^2+^ added against the molar ratio of Zn^2+^ to tau-R3. The data (solid square) was best fitted to a model with a single set of identical sites, and the solid line represents the best fit. (**B**) Spectrophotometric titration of Zn^2+^ to tau-R3, Zn^2+^/tau-R3 ratio ranges from 0–0.5.

**Figure 3 ijms-20-00487-f003:**
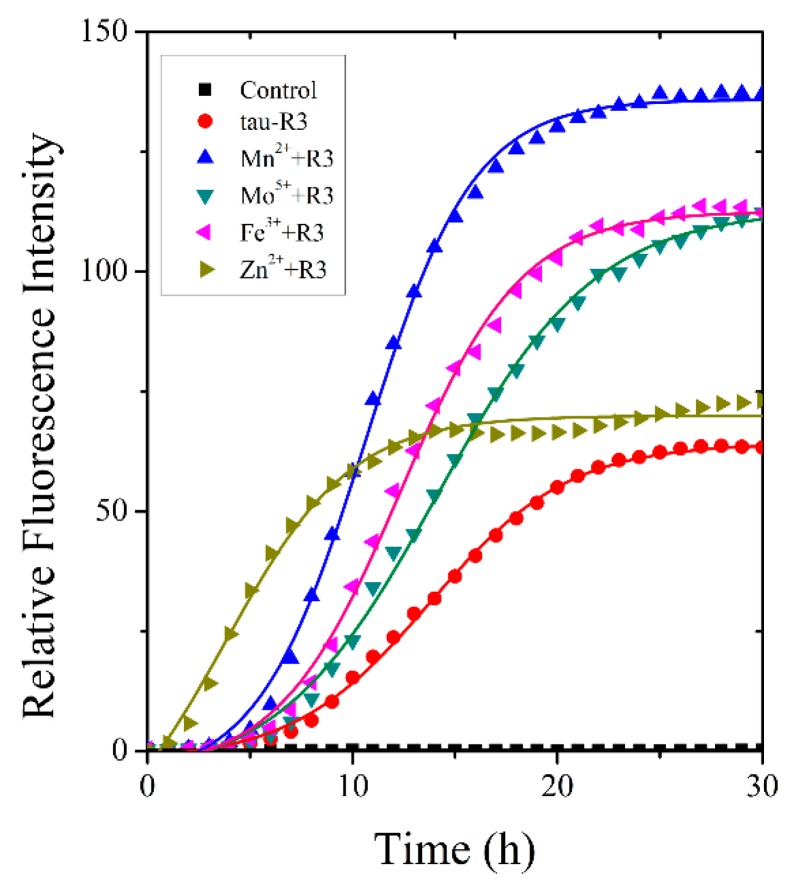
Metal ions altered tau-R3 fibrillization kinetics. 150-μM tau-R3 was incubated with 10 μM of cation (Mn^2+^, Mo^5+^, Fe^3+^ or Zn^2+^) or no cation. The buffer used was 50 mM of Tris-HCl buffer (50 mM Tris-base + 100 mM of NaCl, pH 7.4) containing 16 μM of heparin, and 20 μM of ThT. Solid lines represent fits of the data to the Boltzmann equation. λex = 440 nm, λem = 485 nm.

**Figure 4 ijms-20-00487-f004:**
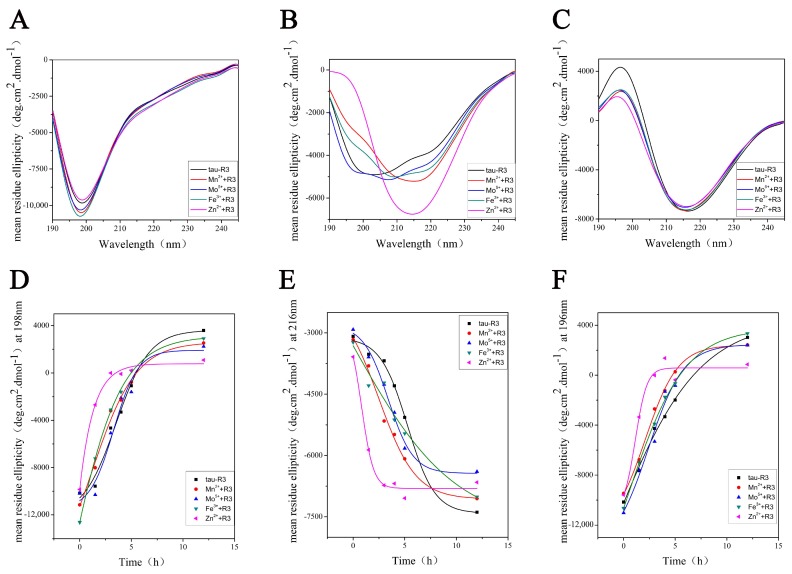
Metal cations altered the secondary structure of tau-R3. (**A**–**C**) Circular dichroism spectra of tau-R3 binding with different additives (Mn^2+^, Mo^5+^, Fe^3+^, and Zn^2+^) incubated at 37 °C for zero, three, and 24 h, respectively. (**D**–**F**) mean residue ellipticity (deg.cm^2^.dmol^−1^) of tau-R3 incubated with different additives (Mn^2+^, Mo^5+^, Fe^3+^, and Zn^2+^) at 198 nm, 261 nm, and 196 nm, respectively in Phosphate buffer (PB) buffer (1.44 g/L Na^2^HPO_4_, 0.24 g/L KH_2_PO_4_, pH = 7.4).

**Figure 5 ijms-20-00487-f005:**
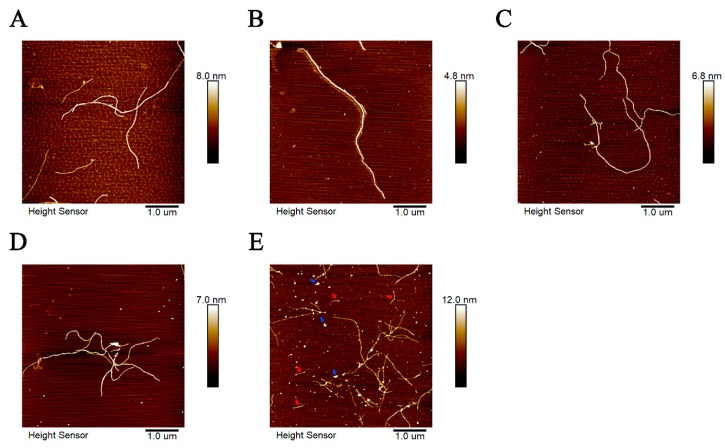
Atomic force microscope (AFM) images of tau-R3 aggregations formed with or without cations ((**A**) tau-R3; (**B**) Mn^2+^+R3; (**C**) Mo^5+^+R3; (**D**) Fe^3+^+R3; and (**E**) Zn^2+^+R3). Tau-R3 (50 μM) was incubated with heparin (16 μM) in the presence or absence of metal ions (10 μM) at 37 °C for 24 h. Then, each sample was diluted 50 times. Shorter filaments were indicated with red arrows and oligomers with blue arrows. The scale bars represent one μm.

**Figure 6 ijms-20-00487-f006:**
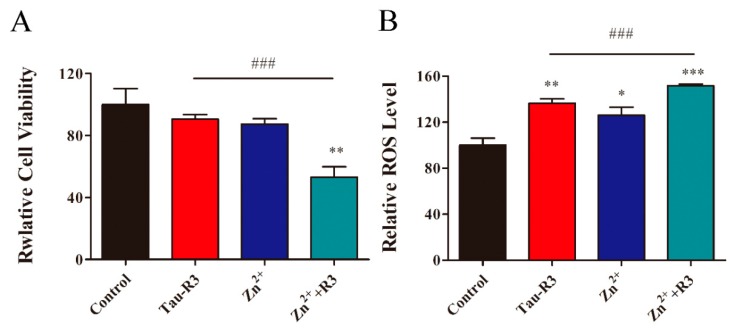
(**A**) Cell viability of Neuro-2A (N2A) cells tested by Cell-Counting Kit-8 Assay upon treatment with: lane 1, the control experiment; lane 2, tau-R3 aggregates; lane 3, Zn^2+^; and lane 4, Zn^2+^-tau-R3 aggregates. (**B**) The reactive oxygen species (ROS) levels in N2A cells upon treatment with: lane 1, the control experiment; lane 2, tau-R3; lane 3, Zn^2+^; and lane 4, Zn^2+^ plus tau-R3. ROS in living N2A cells were monitored by a fluorescence assay kit consisting of 2′,7′-dichlorofluorescin diacetate and quantified with flow cytometry. The final concentrations of tau-R3 and Zn^2+^ are 15 μM and 7 μM. (* *p* < 0.05, ** *p* < 0.01, *** *p* < 0.001 vs. control group; ### *p* < 0.001 vs. tau-R3 treated group).

**Figure 7 ijms-20-00487-f007:**
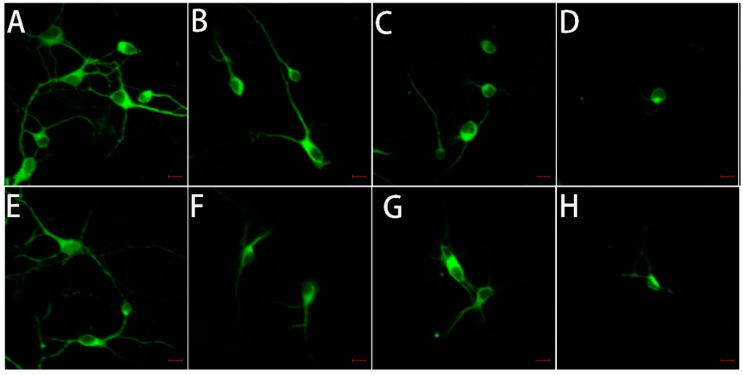
Typical confocal images of primary neurons treated with tau-R3 aggregates, Zn^2+^, or Zn^2+^-tau-R3 aggregates at 37 °C for 12 h. The final concentrations of tau-R3 and Zn^2+^ were 7.5 μM and 3.5 μM. (**A**–**D**) Hippocampal neuron, (**E**–**H**) cortical neuron, (**A**,**E**) the control group, (**B**,**F**) tau-R3 treated neurons, (**C**,**G**) Zn^2+^ treated neurons, (**D**,**H**) Zn^2+^-tau-R3 treated neurons. The scale bars represent 10 μm.

**Figure 8 ijms-20-00487-f008:**
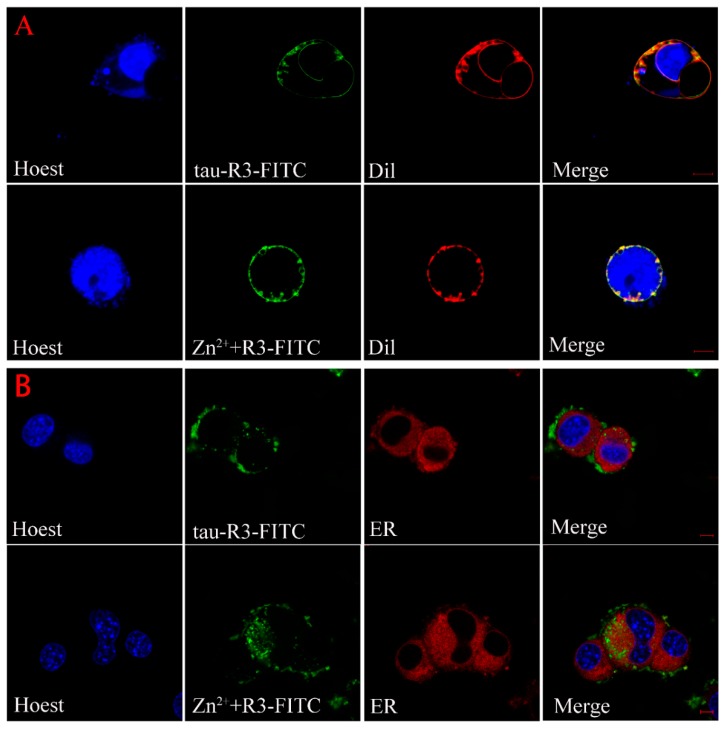
Cellular localization of tau-R3 and Zn^2+^-tau-R3 in N2A cells. The final concentrations of tau-R3 and Zn^2+^ are 7.5 μM and 3.5 μM. Nuclei stained with Hoechst, plasma membrane stained with Dil-Tracker Red (**A**), endoplasmic reticulum stained with ER-Tracker Red (**B**), and Golgi complex stained with Golgi-Tracker Red (**C**). The scale bars represent 5 μm.

**Figure 9 ijms-20-00487-f009:**
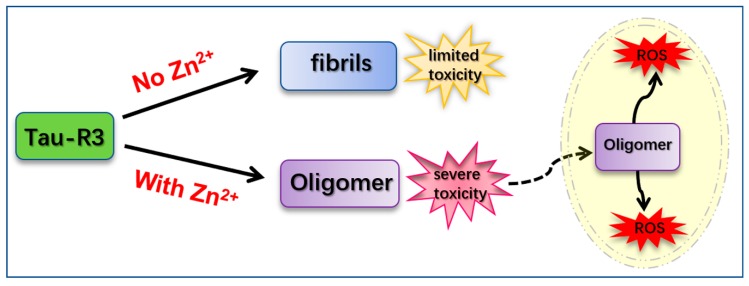
Hypothetical model. The solid arrows represent the process of generation, and the virtual arrow represents the process of entering the cell.

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
