# Peer review of "Zn2+ Aggravates Tau Aggregation and Neurotoxicity"

_ijms, 2019, doi:10.3390/ijms20030487_

Reviewer 1 Report

Li et al. show the cysteine residue of the R3 fragment of the tau protein can bind Zn2+ and other transition metals with low micromolar affinity, extending previous work on the subject.1 Binding of zinc was associated with accelerated aggregation, a shift to shorter filaments and non-fibrillar oligomers, higher cellular uptake, and enhanced ROS production and neurotoxicity2 in N2A cells. The experiments are well designed and support the main conclusions of the manuscript; I have no problem with this aspect.

My only comments relate to the extrapolation of the results to the full-length tau protein. Some research has been recently performed in this respect and should be referenced and discussed at least briefly. In particular, full length tau seems to have a different binding mode (n=1),2, 3 a higher affinity binding site (Ka~1e6),2, 3 and possibly a different aggregation pathway than the R3 fragment. Possible differences between R3 and full-length tau should be discussed.  It would also be helpful to have a very brief discussion on the relevant in vivo concentrations of zinc and possibly copper in the context of the binding affinity.

[1] Jiji, A. C., Arshad, A., Dhanya, S. R., Shabana, P. S., Mehjubin, C. K., and Vijayan, V. (2017) Zn(2+) Interrupts R4-R3 Association Leading to Accelerated Aggregation of Tau Protein, Chemistry 23, 16976-16979.

[2] Hu, J. Y., Zhang, D. L., Liu, X. L., Li, X. S., Cheng, X. Q., Chen, J., Du, H. N., and Liang, Y. (2017) Pathological concentration of zinc dramatically accelerates abnormal aggregation of full-length human Tau and thereby significantly increases Tau toxicity in neuronal cells, Biochim Biophys Acta Mol Basis Dis 1863, 414-427.

[3] Roman, A. Y., Devred, F., Byrne, D., La Rocca, R., Ninkina, N. N., Peyrot, V., and Tsvetkov, P. O. (2018) Zinc Induces Temperature-Dependent Reversible Self-Assembly of Tau, J Mol Biol.

Author Response

Response: The discussion and references have been incorporated into the revised manuscript in red.

1) Physiological concentration of free Zn2+ in cells is between 1 nM and 10 nM, but in AD brains, Zn2+ is enriched and its intracellular concentrations can be in the range of 10-300 μM. [1] Therefore, the binding constant obtained in this study is reasonable. (lines 10-13, Page 10 in the revised manuscript)

(1) Sensi, S. L.et al, Zinc in the physiology and pathology of the CNS. Nat Rev Neurosci 2009, 10, (11), 780-91.

2) Full length tau has two Cys (C291 and C322) residues, which were located in R2 and R3 fragments, respectively. Zn2+ bound to full length tau by interacting with C291 and C322, with the stoichiometry value n = 1 and binding constant exceeding 106 M-1. [2, 3] When Cys291 was mutated to Ala, the binding number and dissociation constant were decreased to 0.44 and 9.71 μM, respectively, [4] which were much comparable with the values obtained with tau-R3 in this study. In addition, Zn2+ also accelerated the fibrillization of full length tau and induced tau to form shorter fibrillar aggregates. [2, 3] ( lines 13-18 & lines 25-26, Page 10 in the revised manuscript

(2) Roman, A. Y.et al., Zinc Induces Temperature-Dependent Reversible Self-Assembly of Tau. Journal of molecular biology 2018.

(3) Hu, J. Y.et al., Pathological concentration of zinc dramatically accelerates abnormal aggregation of full-length human Tau and thereby significantly increases Tau toxicity in neuronal cells. Biochimica et biophysica acta. 2017, 1863, (2), 414-427.

(4) Mo, Z. Y. et al. Low micromolar zinc accelerates the fibrillization of human tau via bridging of Cys-291 and Cys-322. J Biol Chem 2009, 284, (50), 34648-57.

Reviewer 2 Report

Interesting and well documented experimental study demonstrating that Zn2+ can aggravate tau-R3 aggregation and neurotoxicity, ading further clues to understand the relationship between Zn2+ dyshomeostasis and AD etiology.

As far as I can consider, the paper is original, based on personal experimental studies of the authors showing the influence of ZN2+ on 3R tau morphology, its toxicity to neurons, thus presenting a hypothecial model how Zn2+ aggravates 3R-tau aggregation and toxicity, suggesting an involvement of Zn2+ in the pathogenesis of AD.

Additional reference could be given to the effect of Zn on Aß aggregation in its interaction with tau protein (Beyer N et al, JAlzheimers Dis 2012;29:863-73; Jellinger KA, Int Rev Neurobiol 2013;110:1-47).

Author Response

Response:

1) A necessary reference recommended by the reviewer was added at the end of the sentence “The dynamic imbalance of metal ions such as Cu2+, Fe3+, Zn2+, and Ca2+ in the brain is closely related to the pathogenic mechanism of AD”. (Jellinger KA, Int Rev Neurobiol 2013;110:1-47) (line 13Page 2 in the revised manuscript)

       2) According to reviewer’s comments, some introduction and a reference was added! It was found that the mRNA levels of Zn2+ transporters including LIV1, ZIP1, ZnT1, ZnT4, and ZnT6 were increased in the cortex of postmortem brain tissues from AD patients, increasing the likelihood of interactions between Zn2+ and Aβ or tau protein in the brains. (Beyer, N.; Coulson, D. T.; Heggarty, S.; Ravid, R.; Hellemans, J.; Irvine, G. B.; Johnston, J. A., Zinc transporter mRNA levels in Alzheimer's disease postmortem brain. Journal of Alzheimer's disease : JAD 2012, 29, (4), 863-73.) (lines 26-29Page 2 in the revised manuscript)